# IL-6, NLR, and SII Markers and Their Relation with Alterations in CD8+ T-Lymphocyte Subpopulations in Patients Treated for Lung Adenocarcinoma

**DOI:** 10.3390/biology9110376

**Published:** 2020-11-05

**Authors:** Lorenzo Islas-Vazquez, Dolores Aguilar-Cazares, Miriam Galicia-Velasco, Uriel Rumbo-Nava, Manuel Meneses-Flores, Cesar Luna-Rivero, Jose Sullivan Lopez-Gonzalez

**Affiliations:** 1Lung Cancer Laboratory, Department of Chronic-Degenerative Diseases, Instituto Nacional de Enfermedades Respiratorias Ismael Cosío Villegas, Calzada de Tlalpan 4502, Col Seccion XVl, Mexico City CP 14080, Mexico; lorenzo.islasv@gmail.com (L.I.-V.); or daguilarc@iner.gob.mx (D.A.-C.); miriam_galicia_velasco@yahoo.com.mx (M.G.-V.); 2Neumo-Oncology Clinic, Instituto Nacional de Enfermedades Respiratorias Ismael Cosío Villegas, Calzada de Tlalpan 4502, Col Seccion XVl, Mexico City CP 14080, Mexico; urumbo@gmail.com or; 3Posgrado en Ciencias Médicas, Odontológicas y de la Salud, Universidad Nacional Autónoma de México, Avenida Ciudad Universitaria 3000, Mexico City CP 04510, Mexico; 4Department of Pathology, Instituto Nacional de Enfermedades Respiratorias Ismael Cosío Villegas, Calzada de Tlalpan 4502, Col Seccion XVl, Mexico City CP 14080, Mexico; manuelmeneses707@gmail.com (M.M.-F.); lunarivero@hotmail.com (C.L.-R.)

**Keywords:** NLR, SII, IL-6, HMGB-1, TGF-β, lung adenocarcinoma, CD4+ and CD8+ T-lymphocyte populations, naïve, memory, and effector subpopulations, predictive markers

## Abstract

**Simple Summary:**

Lung cancer is the leading cause of cancer mortality worldwide. The most frequent group of lung cancer is the nonsmall cell lung carcinoma. The immune system of cancer patients participates in the crosstalk between inflammatory immune and nonimmune cells and cancer cells. This event is mediated by several molecules called cytokines. Lung cancer patients are frequently diagnostic at advanced stages, so chemotherapy is the major strategy for treatment. Various inflammatory factors have been described as prognostic biomarkers, such as the IL-6 cytokine, neutrophil–lymphocyte ratio (NLR), and systemic immune-inflammation index (SII). In a cohort of patients with lung adenocarcinoma treated with conventional chemotherapy, changes in pro- and anti-inflammatory cytokines, mainly IL-6, the NLR, and SII, were studied. In addition, variations in the percentages of CD4+ and CD8+ T-lymphocyte subpopulations were investigated. Compared to healthy subjects, high levels of IL-6 were detected in patients prior to treatment. In the treated patient group with higher overall survival (OS), this cytokine decreased. Decreases in the NLR and SII values were detected from the third cycle of chemotherapy. Patients with lower OS had significantly lower CD8+ T-lymphocytes and its effector subpopulation. These parameters could be useful as predictive markers in lung adenocarcinoma.

**Abstract:**

Cytokines, key contributors to tumorigenesis, are mediators between inflammatory immune or nonimmune and cancer cells. Here, IL-6 production by tumor cells was assessed in a cohort of patients with lung adenocarcinoma treated with conventional therapy. IL-6 levels and neutrophil–lymphocyte ratio (NLR) or systemic immune-inflammation index (SII) markers were evaluated. Changes in pro- and anti-inflammatory cytokines, HMGB1 concentration, and CD4+ and CD8+ T-lymphocyte populations and their subpopulations were investigated. IL-6 expression was detected immunohistochemically in lung adenocarcinoma biopsies. Cytokines were quantified using the cytometric bead array, and TGF-β and HMGB-1 through ELISA. Clinical parameters were collected to assess NLR and SII. CD4+ and CD8+ T-lymphocytes and naïve, memory, and effector subpopulations were quantified by flow cytometry. The data obtained were associated with patients’ median overall survival (OS). IL-6 showed the highest increase, probably because the lung adenocarcinoma cells produced IL-6. Patients with higher OS had lower NLR and SII from the third cycle of chemotherapy. Patients with lower OS had significantly lower percentages of CD8+ T-lymphocyte and its effector subpopulations, with a concomitant increase in the naïve subpopulation. This study suggests that in addition to the known inflammatory markers, IL-6, CD8+ T-lymphocytes and their effector and naïve subpopulations could be useful as predictive markers in lung adenocarcinoma.

## 1. Introduction

Lung cancer is the leading cause of cancer-related mortalities worldwide, with a five-year overall survival (OS) rate of 15% in the early stages and less than 5% in the advanced stages of the disease [1,2]. Lung cancer is of two main types, including small cell lung carcinoma (SCLC) and non-SCLC (NSCLC). In the NSCLC group, adenocarcinoma is the most common histological type [3].

Cytokines are key contributors to tumorigenesis, participating in the crosstalk between inflammatory immune and nonimmune cells and cancer cells [4]. Notably, IL-6, a pro-inflammatory cytokine produced by different cell types, including immune cells, endothelial cells, cancer-associated fibroblasts, and tumor cells [5], is an important cytokine associated with chronic inflammation [6]. In patients with several types of cancer, including NSCLC, high serum IL-6 levels are related to tumor stage, size, metastasis, and survival [7,8].

Various inflammatory factors have been described as prognostic biomarkers, such as the neutrophil–lymphocyte ratio (NLR), platelet–lymphocyte ratio (PLR), lymphocyte–monocyte ratio (LMR), C-reactive protein ratio, and systemic immune-inflammation index (SII), which influence survival in patients with cancer [9,10,11,12]. The SII was reported as an independent prognostic marker for patients with NSCLC [11,13].

NSCLC is frequently diagnosed at advanced or metastatic stages and based on the tumor, node, and metastasis (TNM) system. These advanced tumors are grouped in stages IIIb, IIIc, and IV [14]. According to the National Comprehensive Cancer Network (NCCN) Guidelines for nonsmall cell lung cancer [15], NSCLC patients in advanced stages showing the EGFR mutations are treated with epidermal growth factor receptor (EGFR) tyrosine kinase inhibitors as molecular target therapy [16]. However, in those patients without EGFR mutations, the use of platinum-based regimens is recommended [17]. Palliative chemotherapeutic regimens based on cisplatin (CDDP) or carboplatin (CBDCA) combined with paclitaxel, docetaxel, gemcitabine, pemetrexed, or vinorelbine are systemic treatment options. Treatment with platinum doublets is limited to four to six cycles, and each cycle is administered every 21 to 28 days [18]. The eligibility of the numbers of cycles depends on the Eastern Cooperative Oncology Group (ECOG) performance status of the patient [15].

Studies have reported that in patients undergoing cancer treatment with platinum-based compounds or other therapeutic drugs, necrotic cell death or cellular stress causes the redistribution of an array of intracellular molecules, called damage-associated molecular patterns (DAMPs) [19,20,21]. High-mobility group box 1 (HMGB-1), one of the most-studied DAMPs, can bind to TLR-2, TLR-4, and RAGE receptors, inducing inflammation and activating the immune response [22,23]. However, a protumoral activity of HMGB-1 has also been reported [24]. High expression of HMGB-1 in NSCLC, compared to normal lung tissues, has been associated with tumor cell proliferation, migration, and angiogenesis [25].

This study was performed in a cohort of patients with lung adenocarcinoma, showing wild-type EGFR, treated with conventional chemotherapy. We investigated whether plasma IL-6 concentration correlates with the NLR and SII values and the expression of IL-6 was evaluated in biopsies of adenocarcinoma patients. We quantified changes in circulating pro- and anti-inflammatory cytokines in transforming growth factor-β (TGF-β) and HMGB-1 concentrations. Moreover, the percentages of CD4+ and CD8+ T-lymphocyte populations and naïve, memory, and effector subpopulations were quantified. Finally, the data obtained were related to the patient’s median OS.

## 2. Materials and Methods

### 2.1. Population Studied

This study included 53 patients with primary lung adenocarcinoma who visited the pneumo-oncology department at the Instituto Nacional de Enfermedades Respiratorias Ismael Cosio Villegas. Lung adenocarcinoma was diagnosed by histological or cytological examination of biopsied material or malignant cells in pleural effusions by two pathologists. This histologic type was identified by morphologic patterns and the markers thyroid transcription factor-1 (TTF-1) and napsin A [17]. The EGFR mutation status was detected in the patients’ biopsies using the Idylla EGFR mutation assay from Biocartis (NV, Mechelen, Belgium). All these procedures are routinely done in the pathology department.

The clinical staging of patients with adenocarcinoma was evaluated using the TNM score [14]. Because chemotherapy is a crucial component of lung cancer treatment, in this study, a cohort of chemotherapy-naïve patients with adenocarcinoma in stages IIIb–IV were included. All patients recruited had as inclusion criteria histologically confirmed lung adenocarcinoma and wild-type EGFR, complete clinical and laboratory data, and follow-up.

The performance status of patients was established based on the Eastern Cooperative Oncology Group (ECOG). Additional inclusion criteria included adequate hematological, renal, and hepatic functions. Patients showing comorbidities, such as severe cardiopulmonary dysfunction, uncontrolled arrhythmia, a history of myocardial ischemia, and active infection, were excluded.

During follow-up in the clinic, peripheral blood samples were collected (Appendix A). The OS rate of patients with lung adenocarcinoma was designated as the time elapsed from the date of palliative chemotherapy initiation to the date of patient death. The Kaplan–Meier curve was used and the average median OS for the entire group was obtained. Based on this data, patients were grouped according to shorter (less than 12 months) or longer (more than 12 months) OS (Appendix A). Demographic characteristics of the patients with lung adenocarcinoma are shown in Table 1.

Written informed consent was obtained from all participants before peripheral blood or biopsy collection. The protocol was approved by the Committee of Science and Bioethics of the Instituto Nacional de Enfermedades Respiratorias Ismael Cosio Villegas.

### 2.2. Expression of IL-6 in Lung Adenocarcinoma Biopsies

After the diagnosis of lung adenocarcinoma was established, residual biological material was collected to detect IL-6 in tumor cells. Briefly, tissue sections were deparaffinized, rehydrated, and treated with citrate buffer 0.01 M, pH 6.0, at 110 °C for 15 min in a NxGen decloaking chamber (Biocare Medical, Pacheco, CA, USA) for heat-induced epitope retrieval. The tissue sections were treated with 3% hydrogen peroxide and 2% mouse serum for blocking endogenous peroxidase activity and nonspecific binding, respectively.

For IL-6 staining, the slices were incubated with rabbit antihuman IL-6 polyclonal antibody (1:50 dilution, #BS-4587R) from Bioss Inc., Boston, MA, USA. Negative (tumor biopsy without a primary antibody) and positive (liver biopsy) controls were used in each experiment (Appendix A). Slices were incubated overnight at 4 °C, washed, and incubated with a biotin-labeled secondary antibody (1:300 dilution, #GTX27055) from Genetex (Irving, CA, USA) at 32 °C for 90 min. Thereafter, the StreptABComplex/HRP (Burlingame, CA, USA) was added and color was developed after incubation with 3,3′-diaminobenzidine (DAB) chromogen/HRP substrate for 10 min. The development of a brown color indicated a positive reaction. Tissue sections were counterstained with Harris hematoxylin and mounted. The intensity of marker immunoreactivity was determined as grade 0 = none, grade 1 = low, grade 2 = moderate, and grade 3 = strong. Microscopy images were captured using the DFC425 C color camera (Leica Microsystem Inc., Wetzlar, Germany) coupled to a Leica DMLB light microscope and the Leica Application Systems v. 3.6.0 software (Leica Microsystem Inc.).

### 2.3. Blood Sample Collection

For the cytokine array, TGF-β, HMGB-1, CD4+, and CD8+ T-lymphocyte subpopulations and peripheral blood samples were collected from 53 patients during the follow-up period, just before the first (baseline), third, and sixth cycles of platinum-based doublet chemotherapy. The regimens are shown in Table 1.

A total of 10–12 mL of blood was obtained through venipuncture. Plasma was collected and immediately frozen at −80 °C until the cytometric bead array (CBA) Human Th1/Th2/Th17 cytokine assay or ELISA cytokine assays were performed. For obtaining peripheral blood mononuclear cells (PBMCs), the blood samples were diluted and centrifuged using a density gradient medium (Lymphoprep, from Axis-Shield, Dundee, Scotland, UK) at 1600× *g* rpm. PBMCs were collected, washed, and stained for multiparametric flow cytometry and CD4+ and CD8+ T-lymphocyte populations and their subpopulations were analyzed.

### 2.4. Quantification of Th1/Th2/Th17 Cytokines Using Cytometric Bead Array

For evaluating changes in cytokine levels in patients with adenocarcinoma during treatment, the Human Th1/Th2/Th17 Cytokine Kit (#560484, BD Bioscience, San Jose, CA, USA) was used according to the manufacturer’ instructions. IL-2, IL-4, IL-6, IL-10, TNF-α, IFN-γ, and IL-17A were quantified. Data were acquired through cytometry using the FACSCanto II and were analyzed using the FCAPArray software v. 3.0 (Softflow, Pécs, Hungary).

### 2.5. Quantification of TGF-β and HMGB-1 by Enzyme-Linked Immunosorbent Assays (ELISA)

Using plasma collected from patients, TGF-β1 was quantified using the Quantikine ELISA Human TGF-β1 Immunoassay (#SB100B) from R&D. For HMGB-1, the human HMGB-1 ELISA Kit (#ST51011) from IBL International (Hamburg, Germany) was used. All assays were performed according to the manufacturer’s instructions. To avoid interassay variations, samples of the same patient were thawed and immediately ran using the same ELISA plate. The optical density was measured at 450 nm in the Multiskan Ascent spectrophotometer (Thermo Scientific, Waltham, MA, USA). Each sample was run in triplicate and the values were averaged.

### 2.6. Data Collection from Cancer Patients

Blood was collected from patients with adenocarcinoma within one week before clinical evaluation by the oncologist and before the chemotherapy cycle. The clinical laboratory data, including routine blood tests and the total number of neutrophils, lymphocytes, and platelets, were obtained from the complete blood count data collected from the medical records. The neutrophil–lymphocyte ratio (NLR) was defined as the proportion of absolute neutrophil count to absolute lymphocyte count. The systemic immune-inflammation index (SII) was calculated using the formula SII = (P × N)/L, where P, N, and L refer to peripheral platelet, neutrophil, and lymphocyte counts, respectively.

### 2.7. Panel of Antibodies for Phenotyping of T-Lymphocyte Subpopulation

The following labeled antihuman monoclonal antibodies (MoAbs) were used: The PE-CF594-conjugated anti-CD3 (UCHT1 clone, #562280) purchased from BD Horizon TM (San Jose, CA, USA), Alexa Fluor 700-coupled anti-CD4 (RPA-T4 clone, #557922) from BD Pharmingen TM (San Jose, CA, USA), and APC/Cy7 anti-CD8 (HIT8a clone, #300926) from BioLegend (San Diego, CA, USA). FITC anti-T-bet (4B10 clone, #644812) and PE anti-CRTH2 (BM16 clone, #350106) from BioLegend, PE anti-RORγ (AFKIS-9 clone, #12-6988) from eBioscience (Waltham, MA, USA), PE-Cy5 anti-CD25 (M-A251 clone, #555433) from BD Pharmingen, and Alexa Fluor 647 anti-CD127 (A019D5 clone, #351318) from BioLegend were used. Effector molecules of CD8+ T-cells were identified using the FITC antigranzyme (GB11 clone, #515403) and antiperforin (B-D48 clone, #353304) MoAbs purchased from BioLegend. PE anti-CD45RO (UCHL1 clone, #304206) and PerCP/Cy5.5 anti-CD27 (0323 clone, #302820) MoAbs from BioLegend were used for quantifying naïve, central, and effector memory and effector T-cells.

### 2.8. Quantification of CD4+ and CD8+ T-Lymphocyte and Their Subpopulations

#### Staining Procedure and Cytometric Analysis

Briefly, 2 × 10^5^ PBMCs were placed in a cytometric tube, centrifuged, and resuspended in 100 μL of PBS/BSA buffer. The cells were incubated with anti-CD3, anti-CD4, anti-CD8, anti-CRTH2, anti-CD25, anti-CD127, anti-CD45RO, and anti-CD27 MoAbs with shaking at room temperature for 30 min, washed, and centrifuged. The cell pellet was treated with 200 μL of Fix/Perm solution (BioLegend) for 20 min. The cells were incubated with anti-T-bet, anti-RORγ, antigranzyme B, and antiperforin MoAbs for 30 min, centrifuged, and fixed with 300 μL of 1% paraformaldehyde.

Events were acquired with the FACSCanto II cytometer, using the BD FacsDiva software. Cells were gated in an FSC-A vs. FSC-H dot plot to exclude doublets. From this population, the lymphocyte region was selected using an FSC vs. SSC dot plot. From this gated population, an SSC-A vs. CD3 dot plot was made. A total of 20,000 events of this population were acquired. From the gated-CD3 region, a CD4 vs. CD8 plot was used to identify and quantify the corresponding T-cell populations. From the CD4+ T-lymphocyte population, Th1, Th2, Th17, and Treg subpopulations were evaluated. From the CD8+ T-lymphocyte population, percentages of CD8+ T-lymphocytes expressing granzyme and perforin (termed cytotoxic T-lymphocytes (CTLs)) were detected. Moreover, from each CD4+ or CD8+ T-lymphocyte population, the corresponding naïve, effector, and central/effector memory subpopulations were quantified.

### 2.9. Statistical Analysis

For normal data distribution, the Shapiro–Wilk test was used. Statistical analysis was performed using the Mann–Whitney test for the intergroup comparison. The Friedman test with Dunn’s multiple comparisons for paired samples was used throughout the treatment. Median OS was defined using the Kaplan–Meier curve. For the correlation between IL-6 and NRL or IL-6 and SII, the Spearman correlation test was employed. Data from IL-6, TGF-β, HMGB-1, NLR, and SII are expressed as median and interquartile range. Data from T-lymphocyte subpopulations are expressed as mean ± standard error of mean (SEM). Analyses were performed using the statistical program Graph Pad Prism 8 (GraphPad Software, La Jolla, CA, USA). Values of *p* < 0.05 were considered statistically significant.

## 3. Results

### 3.1. Baseline Patient Characteristics

Demographic and clinicopathological features of 53 patients with lung adenocarcinoma enrolled in the study and characteristics grouped by OS are shown in Table 1. Patients included in the longer OS (>12 months) group were older compared to the shorter OS (<12 months) group. Moreover, a significant decrease in the percentage of CD8+ T-lymphocytes was detected in the shorter OS group compared to the longer OS group. Other demographic characteristics, including treatment regimens and clinicopathological parameters, did not differ significantly between the shorter and longer OS groups. For comparison, data from patients at 3rd and 6th cycle of treatment are shown in Appendix A.

### 3.2. IL-6 Staining in Lung Adenocarcinoma Biopsies

Staining for IL-6 was performed in biopsies collected from 43 patients to detect whether lung adenocarcinoma tumors, in addition to stromal cells, are a supplementary source of IL-6. Based on adenocarcinoma histological subtype, 18 were solid, 12 acinar, 6 lepidic, 5 papillary, and 2 micropapillary. IL-6 was detected in the cytosol of tumors with a homogeneous distribution in 90% of the cases. Strong staining was observed in 72% of them. IL-6 expression was independent of the adenocarcinoma subtype (see Figure 1 and Table 2).

### 3.3. Quantification of Cytokines, TGF-β, and HMGB-1 in the Plasma from Patients with Lung Adenocarcinoma

In a previous report, we quantified seven cytokines and TGF-β in the plasma of untreated patients with lung adenocarcinoma. A statistically significant increase in IL-2, IL-4, IL-6, and IL-10 levels compared to the control group was found. Notably, IL-4 and TGF-β increased two-fold and IL-10 increased three-fold, whereas IL-6 showed the highest increase of approximately seven-fold [26].

In the present study, changes in the aforementioned cytokines TGF-β and HMGB-1 were evaluated during follow-up. In general, levels of cytokines studied remained unchanged (Appendix A). Only circulating IL-6 levels decreased significantly through the treatment and no changes were detected in TGF-β and HMGB-1 plasma concentrations (see Figure 2).

When IL-6 levels were analyzed considering the patients’ median OS, no significant changes were found in the lower OS group. In contrast, in the higher OS group, a significant decrease of IL-6 was detected from the third treatment cycle and this decrease was maintained in the sixth treatment cycle.

The median of the TGF-β level detected was 20.5 ng/mL, which remained unchanged as treatment progressed. Similar circulating concentrations were detected irrespective of the OS groups. For HMGB-1, the median concentration found was 1.7 ng/mL, which remained unchanged as the treatment progressed (see Figure 3).

### 3.4. NLR and SII Values in Patients with Lung Adenocarcinoma

The NLR and SII values were evaluated in patients with adenocarcinoma as treatment progressed. At baseline, the median values of NLR and SII were 3.9 and 1104, respectively. In the third cycle, these values decreased significantly by approximately 50%. Despite a slight increase in NLR in the sixth cycle being detected, this value was significantly lower than the baseline (see Figure 4).

When these data were associated with median OS and compared between lower and higher OS groups, the patient group with lower OS had a significant decrease of less than 50% in NLR and SII values in the third cycle; however, these values returned to the baseline in the sixth cycle. In contrast, in patients with higher OS, a significant decrease of 50% in both inflammatory markers was detected in the third cycle, and this decrease was maintained in the sixth treatment cycle (see Figure 5).

### 3.5. Correlation of NLR and SII Markers with IL-6

Previous reports indicate that neutrophils play a key role in modifying NLR and SII [27]; therefore, the correlation between IL-6 and NLR or SII was evaluated. Positive correlations were found in both cases (see Figure 6).

### 3.6. Percentages of CD4+ T-Lymphocytes and Their Subpopulations

Inflammation is a hallmark of cancer and local and systemic inflammatory cells affect tumor development and progression [28]. As in the higher OS group, significant decreases of IL-6 were detected since the third cycle compared to IL-6 levels detected before treatment. We evaluated whether the cytokine environment, even at advanced stages of cancer, could influence the percentages of CD4+ T-lymphocyte subpopulations, in particular Th1 (CD4+ T-bet+), Th2 (CD4+ CRTH2+), Th17 (CD4+ RORγ+ cells), and Treg (CD4+CD25+CD127-) subpopulations. Based on median OS, in patients with higher OS, a nonsignificant increase of approximately two-fold in the Th1 subpopulation, compared to patients with lower OS, was detected (see Figure 7).

### 3.7. Percentages of CD8+ T-Lymphocytes and Cytotoxic T-Lymphocytes (CTLs) Expressing the Cytolytic Machinery

Tumor cells evade the host’s immune response using several mechanisms [29,30]. Reports indicate that some anti-inflammatory cytokines or growth factors released by tumor cells negatively impact the proportion of CD8+ T-cells or inhibit the synthesis of cytolytic molecules [31]. The proportion of CD8+ T-lymphocytes in the patient group reached less than 30%. However, when these cells were quantified according to the patients’ median OS, the percentages of CD8+ T-lymphocytes in the lower OS group reached approximately 20% of total CD3+ T-lymphocytes. In contrast, in the high OS group, this value was more than 30%. The values were maintained at the follow-up and the differences between them were statistically significant.

In addition, the percentages of CD8+ T-lymphocytes expressing granzyme and perforin (CTLs) were quantified. A total of 40% of CD8+ T-lymphocytes expressed granzyme and perforin throughout the treatment period was found. However, when this population was evaluated according to median OS, patients with lower OS, compared to those with higher OS, showed a significant decrease in systemic CTLs. This decrease was detected as the baseline (before application of the first treatment cycle) but reached statistical significance in the sixth treatment cycle (see Figure 8).

### 3.8. Percentages of Naïve, Memory, and Effector Subpopulations from CD4+ and CD8+ T-Lymphocytes

The proportion of naïve, effector, and central/effector memory CD4+ and CD8+ T-lymphocytes were quantified. According to the OS and compared to baseline, the percentages of distinct phenotypes of CD4+ T-lymphocyte subpopulations remained unchanged during treatment. However, in patients with lower OS, a nonsignificant decrease in the proportion of effector and memory/effector cells compared to patients with longer OS was detected. In addition, a concomitant and nonsignificant increase of naïve CD4+ T-cells compared to the longer OS group was found.

Regarding CD8+ T-lymphocytes, a high proportion of cells with effector phenotype was detected in patients with higher OS compared to patients with shorter OS. In contrast, patients with lower OS showed a higher percentage of naïve T-lymphocytes of almost two-fold compared to those of the higher OS. Data reached statistical significance only in the sixth cycle of treatment (see Figure 9).

## 4. Discussion

Cancer-related inflammation, a hallmark of cancer, participates in cancer development and progression. Mounting evidence suggests that systemic inflammation is associated with poor clinical outcomes in patients with solid tumors, including NSCLC [28]. In the tumor microenvironment, cytokines are released by the immune and nonimmune cells and the tumor [4]. In this context, complex interactions occur between the soluble and cellular components [32]. IL-6 is reported to be the most important tumor-promoting cytokine [33]. Studies have detected high concentrations of IL-6 in patients with cancer. Moreover, IL-6 has been related to tumor stage and size, metastasis, and survival [7,8]. Chang et al. [7] reported that in patients with NSCLC who were treated with chemotherapy, high levels of IL-6 were associated with a poor response and survival outcome. Haura et al. [34] reported the expression of both the IL-6 and IL-6 receptors (IL-6R) in NSCLC, suggesting that the autocrine loop promotes lung cancer aggression.

Here, we detected IL-6 in various histologic subtypes of lung adenocarcinoma and found that IL-6 staining intensity was stronger in the tumor than the tumor stromal cells. In addition, IL-6 was synthesized independently of the histological subtype.

We have previously reported that IL-6 in smoking and nonsmoking subjects ranged from 1 to 2 ng/mL. Rose-John [5] reported, in the blood of healthy subjects, levels of IL-6 similar to those found in the report. In the present study, plasma IL-6 levels in patients with adenocarcinoma ranged from 6 to 8 ng/mL, an increase of three or more times. Changes in the IL-6 levels could be detected in lung cancer follow-up and were associated with patient overall survival. Li et al. [33] reported that the exogenous addition of IL-6 activates the Jak/STAT3 pathway, promoting NSCLC cell growth and survival. Based on our results, the histologic lung adenocarcinoma subtypes produce IL-6, and this cytokine may be promoting cell proliferation and tumor progression. However, studies in earlier stages of lung adenocarcinoma are required to support this proposal.

The levels of inflammatory NLR and SII and IL-6 have been independently associated with survival in patients with NSCLC [12,13,35]. However, to our knowledge, no study has evaluated the association between inflammatory markers and IL-6 levels during follow-up in patients with lung adenocarcinoma.

In this study, we found a positive correlation between NLR or SII and IL-6. Moreover, IL-6 could be used as an additional marker for the follow-up of clinical response in chemotherapy, as well as in the most recent options of therapy. Studies of inflammatory markers and IL-6 during the follow-up of the diverse options of treatment are needed. Even though this study focused on lung adenocarcinoma, IL-6 could be useful as a marker of treatment response in other cancers.

In our previous study, we reported increased levels of IL-4, IL-6, IL-10, and TGF-β detected in the peripheral blood of lung adenocarcinoma patients [26]. These cytokines and others have been implicated as growth factors for tumor cells and are also known to inhibit the host–immune response. More studies on the cytokine array system are needed to evaluate the role that they play in the pathogenesis of lung adenocarcinoma or other cancers. In addition, it is required to analyze possible changes during the distinct clinical.

According to the NCCN guidelines, patients with lung adenocarcinoma without EGFR mutations or patients for whom molecular targeted therapy was unsuccessful are candidates for conventional chemotherapy [15]. Studies have revealed that some therapeutic drugs induce cell death, releasing or relocalizing DAMPs. HMGB-1 has been associated with cancer hallmarks, including tumor proliferation, invasion, and angiogenesis. Studies have suggested that HMGB-1 is a prognostic and predictive marker for NSCLC [36]. In patients with progressive NSCLC, high levels of serum HMGB-1 have been linked to shorter OS [37]. We quantified HMGB-1 as a key DAMP release during the follow-up of lung adenocarcinoma patients to evaluate its potential as a biomarker. In our study, no change in HMGB-1 concentrations during treatment could be associated with advanced stages of lung adenocarcinoma. Receptors, including RAGE, TLR2, and TLR4, may be upregulated in the adenocarcinoma or stroma cells. The binding of the soluble HMGB-1 by membrane-bound or soluble receptors could be interfering with the accurate quantification of this DAMP [24,38]. More studies are required to demonstrate these aspects.

During conventional treatment of lung adenocarcinoma, changing tumor microenvironment, including cytokines, chemokines, DAMPs, and growth factors, may alter host–immune cells and distinct, complex tumor microenvironments could be shaped according to the patients’ OS [39]. As inflammatory markers take into consideration the absolute lymphocyte and neutrophil counts, the amount of these immune cells might be affected by the cytokine environment. Therefore, a possible association with the patients’ clinical response could be detected. The inflammatory markers change as treatment progresses. We found that in patients with shorter OS, these markers returned to high levels in the last treatment cycle. In particular, the CD4+ T-lymphocytes and their subpopulations were maintained in similar proportions. Respect to total CD8+ T-lymphocytes and their effector subpopulations, identified by phenotypic markers and cytolytic proteins, decreased. The reduced proportion of these cells was balanced by an increase in CD8+ T-lymphocytes with the naïve phenotype. The altered cellular pattern detected in lung adenocarcinoma patients with shorter OS, with respect to those with longer OS, may be a consequence of distinct immunosuppressive mechanisms induced by tumor cells [29,40]. The tumor aggressiveness and the size, number, and location of the metastases might influence the data obtained. The drugs used in the chemotherapy may affect, in addition to tumor cells, the functional activity of the immune cells but not their proportion.

With the development of platinum-based drug delivery systems, chemotherapy will remain a key component of lung cancer treatment [41]. This early and basic study suggests that the evaluation of inflammatory markers, including IL-6, and quantification of CD8+ T-lymphocytes and their subpopulations can be used to follow-up patients with lung adenocarcinoma. More studies with a larger number of patients are needed to support the data obtained in our study. In addition, studies investigating the role of these parameters in cancer patients treated with molecular targeted therapy or immunotherapy alone or in combination with cytotoxic chemotherapy, are desirable.

## 5. Conclusions

In this study, eight cytokines and HMGB1 were quantified in patients with lung adenocarcinoma. The effect of chemotherapy in the systemic cytokine microenvironment and its influence on inflammatory markers and CD4+ and CD8+ T-lymphocytes and their subpopulations was analyzed. Based on the median OS of patients, two groups were established. In the shorter OS group, the IL-6 concentrations tended to increase as treatment progressed. In contrast, in the longer OS group, IL-6 significantly decreased and this reduction was maintained until the sixth cycle. The circulating inflammatory markers, NLR and SII, were also detected in the cancer groups. A positive correlation was found when each inflammatory marker was compared to IL-6 levels. In addition, CD8+ T-lymphocytes were the primary immune cells affected prior to starting therapy in patients with cancer. In patients with shorter OS, a reduction in the total number of this population and in the CTLs were quantified. This reduction was compensated by an increase in CD8+ T-lymphocytes with the naïve phenotype. This early study suggests that the quantification of inflammatory markers, including IL-6, and CD8+ T-lymphocytes and their subpopulations could be used during follow-up of clinical response in patients with lung adenocarcinoma. Analysis of the proposed markers is required in patients treated with molecular targeted therapy or immunotherapy, alone or in combination with chemotherapy.

## Figures and Tables

**Figure 1 biology-09-00376-f001:**
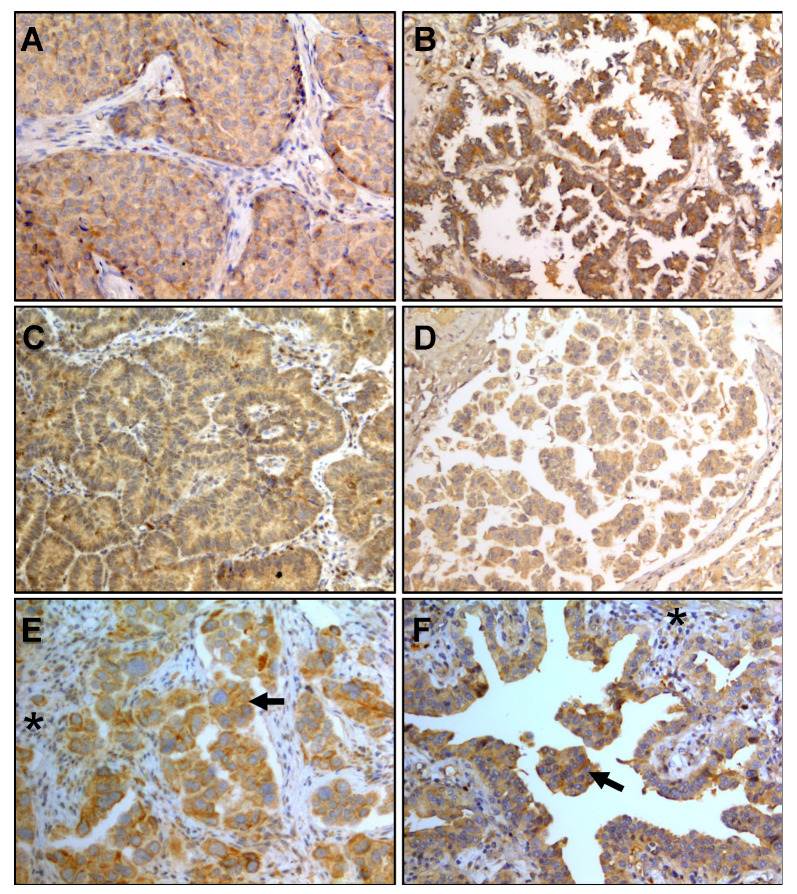
Expression of IL-6 in lung adenocarcinoma histologic subtypes. (**A**) solid, (**B**) lepidic, (**C**) papillary, and (**D**) micropapillary. (**E**,**F**) Comparison between tumor (←) and stroma (*) staining is shown. Magnification: 200×.

**Figure 2 biology-09-00376-f002:**
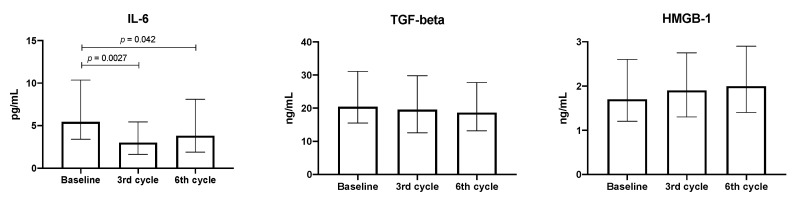
Concentration of plasma IL-6, TGF-β, and HMGB-1 throughout treatment, *n* = 53. Median and interquartile ranges are shown.

**Figure 3 biology-09-00376-f003:**
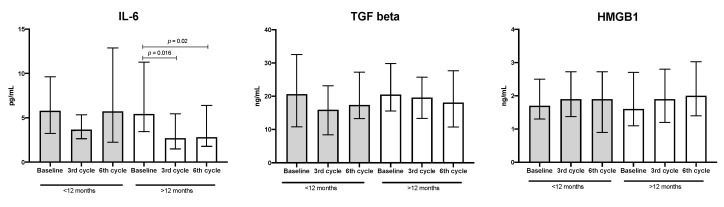
Concentration of IL-6, TGF-β, and HMGB-1 in patients with shorter (<12 months, *n* = 22) or longer (>12 months, *n* = 31) OS. Median and interquartile ranges are shown.

**Figure 4 biology-09-00376-f004:**
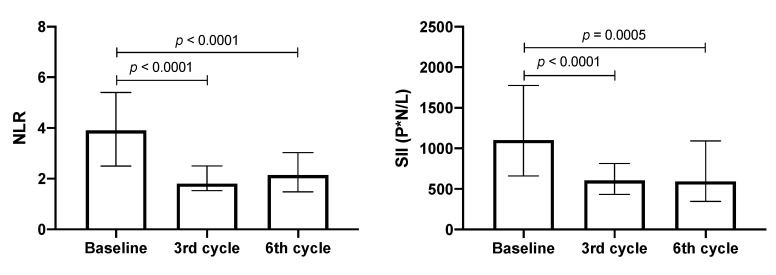
Neutrophil–lymphocyte ratio (NLR) and systemic immune-inflammation index (SII) values during treatment. Median and interquartile ranges are shown (*n* = 53).

**Figure 5 biology-09-00376-f005:**
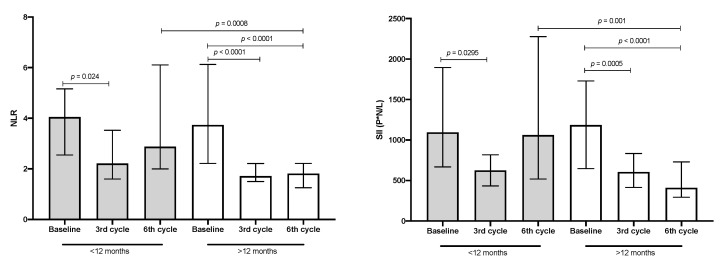
Neutrophil–lymphocyte ratio (NLR) and systemic immune-inflammation index (SII) values. Patients with shorter (<12 months, *n* = 22) and longer (>12 months, *n* = 31) OS. Median and interquartile ranges are shown.

**Figure 6 biology-09-00376-f006:**
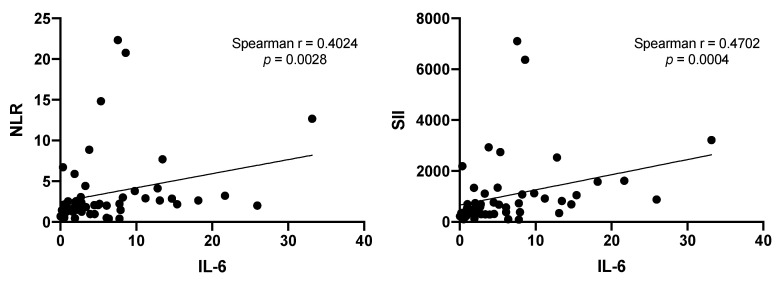
Positive correlations between IL-6 and neutrophil–lymphocyte ratio (NLR) or IL-6 and systemic immune-inflammation index (SII) are shown.

**Figure 7 biology-09-00376-f007:**
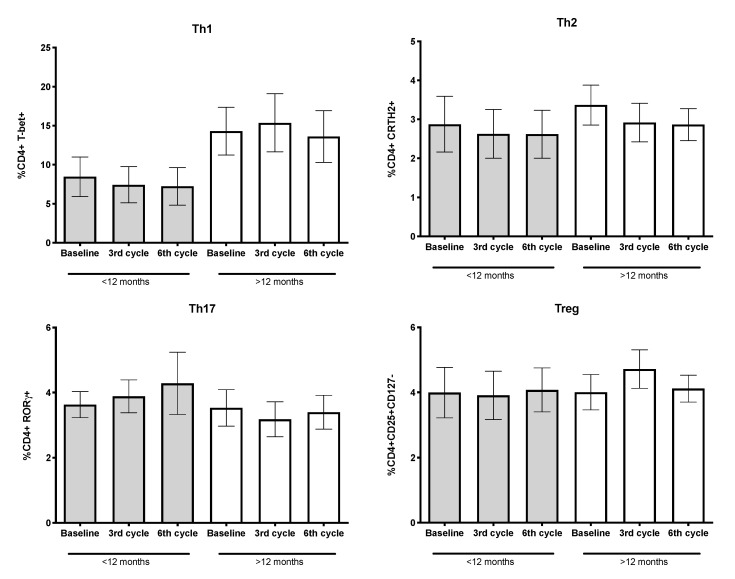
Percentages of CD4+ T-lymphocyte subpopulations. Th1, Th2, Th17, and Treg were quantified by detecting membrane and nuclear markers. Data obtained from patient groups with shorter (<12 months, *n* = 11) and longer (>12 months, *n* = 19) OS is shown. Mean ± SEM are indicated.

**Figure 8 biology-09-00376-f008:**
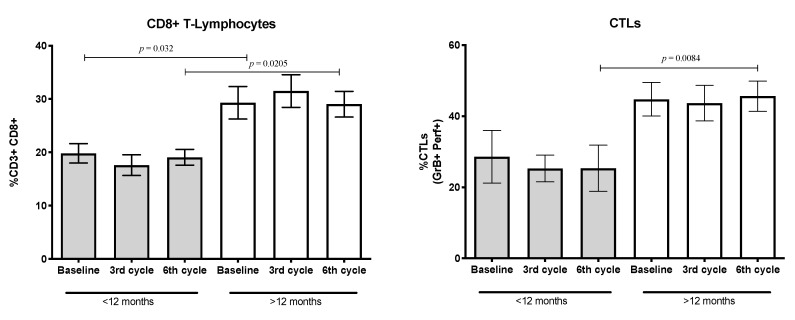
Percentage of CD8+ T-lymphocytes and cytotoxic T-lymphocytes (CTLs) expressing granzyme B and perforin in groups with shorter (<12 months, *n* = 11) and longer (>12 months, *n* = 19) OS. Mean ± SEM are shown.

**Figure 9 biology-09-00376-f009:**
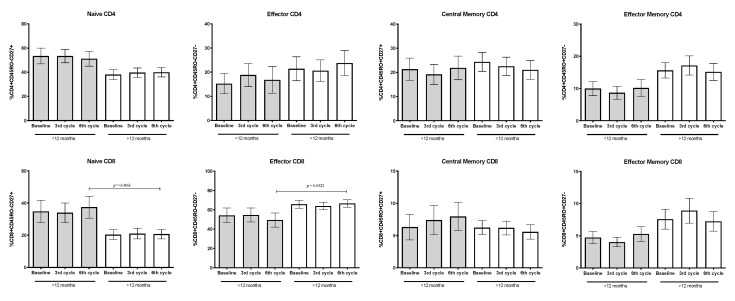
Percentages of naïve, effector and memory CD4+ and CD8+ T-lymphocyte subpopulations. Variations according to treatment cycle in groups with shorter (<12 months, *n* = 11) and longer (>12 months, *n* = 19) OS is shown. Mean ± SEM are shown.

**Table 1 biology-09-00376-t001:** Demographic and baseline characteristics of lung cancer patients.

Characteristic	Total Group	Patients With:		*p*
		Shorter Survival ^1^	Longer Survival ^1^	
n	53	22	31	-
Age (years) ^2^	60 (31–89)	52 (31–75)	61 (37–89)	0.04
Female	22	9	13	-
Male	31	13	18	
Stage				
IIIb-c	13	6	7	-
IV	40	16	24	
ECOG				
1	36	16	23	-
2	17	6	8	
Treatment				
CDDP/Paclitaxel	28	10	18	-
CDDP/Pemetrexed	14	8	6	
CBDCA/Paclitaxel	7	3	4	
CBDCA/Pemetrexed	4	1	3	
Median OS months ^2^	12	9 (5–12)	21 (13–35)	0.0001
Clinical parameters				
Leukocytes (10^3^/mm^3^) ^3^	8.6 (7–11.5)	10.2 (6.9–12.4)	8 (6.9–9.8)	0.0925
Lymphocytes (10^3^/mm^3^) ^3^	1.6 (1.2–2.1)	1.8 (1.3–2.3)	1.6 (1.1–2.1)	0.4379
TGF-β (ng/mL) ^3^	20.5 (15.5–31.1)	20.6 (10.8–32.5)	20.5 (15.5–29.8)	0.9524
HMGB-1 (ng/mL) ^3^	1.7 (1.2–2.6)	1.7 (1.3–2.5)	1.6 (1.1–2.7)	0.7588
IL-6 (pg/mL) ^3^	5.4 (3.4–21.4)	5.8 (3.2–9.6)	5.4 (3.4–11.3)	0.9679
NLR ^3^	3.9 (2.5–5.4)	4.1 (2.5–5.2)	3.7 (2.2–6.1)	0.9608
SII ^3^	1104 (661.7–1777)	1094 (666.9–1894)	1186 (647.5–1730)	>0.999
T-lymphocytes CD4+ (%) ^4^	38.37 ± 3.29	43.40 ± 5.34	35.58 ± 4.13	0.1867
Th1 (%) ^4^	12.21 ± 2.20	8.46 ± 2.53	14.30 ± 3.06	0.1749
Th2 (%) ^4^	3.19 ± 0.41	2.88 ± 0.71	3.37 ± 0.51	0.4289
Th17 (%) ^4^	3.56 ± 0.38	3.63 ± 0.40	3.53 ± 0.56	0.3879
Treg (%) ^4^	4.00 ± 0.44	3.99 ± 0.77	4.01 ± 0.54	0.9063
CD8+ (%) ^4^	25.91 ± 2.22	19.81 ± 1.82	29.30 ± 3.05	0.032
CTLs (GrB+Perf+) (%) ^4^	38.98 ± 4.23	28.59 ± 7.43	44.75 ± 4.73	0.0987

^1^ Patients were categorized according to median overall survival (OS). ^2^ Median (min − max). ^3^ Median (interquartile range). ^4^ Mean ± SEM. ECOG = Eastern Cooperative Oncology Group Performance Status. CDDP = Cisplatin. CBDCA = Carboplatin. NLR = Neutrophil–lymphocyte ratio. SII = Systemic immune-inflammation index. GrB = Granzyme B. Perf = Perforin.

**Table 2 biology-09-00376-t002:** Intensity of IL-6 staining in lung adenocarcinoma subtypes.

Adenocarcinoma Subtype	Total	Intensity			
		Strong	Moderate	Low	None
Solid	18	12	4	0	2
Acinar	12	10	2	0	0
Lepidic	6	5	0	0	1
Papillary	5	3	1	0	1
Micropapillary	2	1	1	0	0

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
