# Peer review of "IL-6, NLR, and SII Markers and Their Relation with Alterations in CD8+ T-Lymphocyte Subpopulations in Patients Treated for Lung Adenocarcinoma"

_biology, 2020, doi:10.3390/biology9110376_

Round 1

Reviewer 1 Report

In the present study by Islas-Vazquez, the authors describe the association of certain cytokines and lymphocyte parameters and their association with survival in a cohort of lung adenocarcinoma patients.

They highlight some interesting associations like an altered CD8+ lymphocyte ratio or IL6 changes in patients with longer median OS. The data, while not ground breaking is certainly interesting. The collection of data at specific time points during chemotherapy is strength of this study. The Figures are nice, and the statistics is adequate.

However, the clinical impact might be limited. Most importantly, the authors describe the association of immune markers at certain time point. This might be interesting but do not help in a clinical course. To define a clinical important biomarker, it is important to define the level of a biomarker with survival but this association was not performed despite having the clinical data present. An association of a biomarker at 12 month of survival is not proving this biomarker to be predictive or prognostic. Interestingly, the authors highlight multiple times the use of Kaplan-Meier Curves but I do not find them in the manuscript. Please clarify this in the manuscript.

Additionally, some points need to be addressed:

  • In the Introduction (line 68-71) the author suddenly speak about EGFR mutated lung cancer. Were the patients in this cohort EGFR positive. Additionally, the justification of chemotherapy was quite bizarre and disturbing. The most recent guidelines define much different criteria for the initiation of chemotherapy than EGFR mutations. Please clarify that in the introduction and refer to the most recent Guidelines of your country and also to the most recent NCCN or ESMO guidelines.
  • In Table 1, please add all p-values instead of mentioning (ns). Eg Th1 looks very different in Figure7 despite being ‘ns’ according to the authors. It is also quite interesting to see a significant p value for the CD8+ ratio but not for the CD4+ ratio despite the fact that those two values are usually inversely correlated.
  • In Line 240, “TGF-beta and HMGB-1 levels were found in nanograms”. This sentence is a bit odd and not providing any scientific information. I would recommend to delete it.
  • In line 332, the authors state that IL6 was stronger in tumor cells than in stromal cells but this data is nowhere shown. I can only find an image with some images of IL6 staining but not the data across all patients for stromal and tumor cell IL6 individually as well as for the different subtypes. Please provide the data or delete this statement.
  • In line 334-337, the authors try to compare IL-6 values across different studies. This should not be performed and only values obtained in the same study can be compared. Please delete this or provide data from an age-matched cohort.
  • In Line 345, the authors discuss the use of IL6 as biomarker in patients treated with TKI or Immunotherapy but do not show any data to support this statement. Please provide data and rational or delete.
  • What is meant with : “This non-invasive marker could provide additional information 345 to conventional follow-up studies.”
  • Generally, the discussion is a bit exaggerating the results and I would snot support all conclusions made by the authors.
  • Some sentences throughout the manuscript are confusion. Please critically revise the manuscript (best with the help of a native speaker or service) to clarify the text.

Reviewer 2 Report

Islas-Vazquez et al. produced a significant body of evidence to demonstrate the utility of immune cell and inflammatory markers in lung adenocarcinoma patients. The authors have characterized expression profiles of IL-6 and other cytokines as well as T-lymphocyte populations in patient samples before and after chemotherapy. They concluded that sustained decrease or increase of IL-6 or CD8+ T-lymphocytes, respectively, may serve as a prognostic biomarker for longer lung cancer patient survival. Altogether, this is a nice work with important clinical consequences for the diagnosis of early stages of lung cancer. The manuscript is well written and conveniently structured into discrete paragraphs relevant for each figure presented. There are only few minor points, which could be further corroborated upon, listed below. Namely, the authors are encouraged to take the opportunity to accompany their article with additional supplementary data figures/tables that would strengthen the overall significance and the impact of their study for the scientific/medical community.

1) Could the authors show how much time has elapsed between the baseline, 3rd, and 6th chemotherapy cycle and what was the variation in this parameter between the patients? This could be plotted in a new supplementary table or figure.

2) The sentence "IL-6 showed the highest increase in peripheral blood previous to treatment but decreased in patients with higher overall survival (OS)" (line 28) does not make sense for the two following reasons:

a) It is not clear to what is the highest increase of IL-6 being compared to? Is it in relation to other cytokine levels, pre-malignant stage, or chemotherapy cycles? Or do the authors mean that the baseline level of IL-6 was highest in peripheral blood? If yes, compared to what?

b) Whereas the first part "IL-6 showed the highest increase in peripheral blood previous to treatment" relates to IL-6 prior to cancer therapy, it is not directly clear whether the second part "but decreased in patients with higher overall survival (OS)" refers to patients that have undergone chemotherapy or not? Please stress this fact in the second half of the sentence.

3) Please change "decrease in the NLR and SII values were" to "decrease in the NLR and SII values was" or "decreases in the NLR and SII values were" (line 29).

4) Please replace "cycle" with "cycle of chemotherapy" (lines 30, 45).

5) The sentence "Patients with lower OS had significantly lower CD8+ T-lymphocyte and effector subpopulations" (line 45) repeats from "Patients with lower OS had significantly lower CD8+ T-30 lymphocyte and effector subpopulations" (line 30).

6) Please change "T-lymphocytes" to "T-lymphocyte" (lines 85, 187, 276, 284, 308).

7) Please provide catalog numbers for all antibodies used.

8) Please change "stablished" to "established" (line 112).

9) It is not clear what the authors mean by "normal serum" (line 116).

10) The authors have mentioned the use of "positive and negative controls (with no primary antibody)" (line 119) for the immunohistochemical staining of IL-6 (Figure 1). Could example images of these be provided next to the tested patient samples in Figure 1 or a new supplementary figure?

11) Please define abbreviation for "DAB" (line 123), "CDDP" (Table 1), "CBDCA" (Table 1), "GrB" (Table 1), "Perf" (Table 1), "IL-6R" (line 329).

12) Please change "immunoractivity" to "immunoreactivity" (line 124).

13) Please replace "CBA" with "cytometric bead array" as it is part of section title (line 139).

14) The authors claim that "IL-2, IL-4, IL-6, IL-10, TNF-alpha, IFN-gamma, and IL-17A were quantified" (line 142), however the results show values only for IL-6 (Table 1, Figures 2,3,6). Please provide data for all other cytokines or remove them from this sentence.

15) The authors clam that they have used the FOXP3 (line 168), PerCP/Cy5.5 (line 171), and FOX3 (line 179) antibodies, however results with these targets are missing from the manuscript.

16) What concentration of paraformaldehyde did the authors use for fixing the cells (line 180)?

17) The abbreviation "CTL" is not intuitively defined (line 188). Does this stand for cytotoxic T lymphocytes?

18) Please change "characteristics of patient" to "patient characteristics" (line 201).

19) Could the authors be more specific about the chemotherapy regimens applied and perhaps provide a brief introduction into lung cancer chemotherapy? Why for example the following drug combinations were used: CDDP/Paclitaxel, CDDP/Pemetrexed, CBDCA/Paclitaxel, CBDCA/Pemetrexed (Table 1)? Please make your article accessible to general audience as not all readers may be lung cancer experts.

20) It is not clear what do the stage (IIIb-IV and ECOG 1/2) data mean in Table 1? Do these refer to lung cancer patient numbers? If yes, why these numbers don't add up to the total patient counts (n)? What does the numerator and denominator actually represent? Please be more transparent in plotting complex tabulated data.

21) It is also not clear what does "ECOG 1/2" mean in Table 1? Please define this term in the table legend.

22) What does "(First-line regimen)" exactly mean in Table 1? Was this regimen same between all the chemotherapy cycles? Please indicate these facts in the table legend.

23) The authors are advised to use the following character "±" instead of underlined "+" in Table 1. Also, would it be please possible to have the individual numbers better aligned horizontally so that they become lined up and centered under each other in each of the Table 1 columns?

24) Please replace "T-lymphocytes (%) CTL’s" with "T-lymphocytes CTLs (%)" in Table 1. Also adjust the font height of this caption so that it consistently matches the size of other captions in Table 1.

25) Please change "18 were solid, 12 acinar, six lepidic, five papillary, and two" to "18 were solid, 12 acinar, 6 lepidic, 5 papillary, and 2" or "eighteen were solid, twelve acinar, six lepidic, five papillary, and two" (line 215).

26) "Strong staining of IL-6 was homogeneous distributed detected" is not grammatically correct (line 216). Please rephrase.

27) Please indicate what does the purple nuclei-like staining represent and why it was included in Figure 1. It would be useful to mention the stain in the Materials and Methods section as well.

28) Would it be possible to include a supplementary table summarizing clinical data following the 3rd and 6th chemotherapy cycle similar to Table 1? Please reassign the "data not shown" reference accordingly (line 229).

29) Please replace "concentration were" with "concentrations were" or "concentration was" (line 241).

30) It is not clear what do "<12 months" and ">12 months" indicators refer to in Figures 3, 5, 7–9? Please indicate this in the graphs or the respective figure legends.

31) 6th cycle is plotted preceding the 3rd cycle in TGF-beta >12 months group (Figure 3). Please correct.

32) Although the authors claim that "At the sixth cycle, a slight increase in NLR was detected, with a statistically significant difference" (line 250), this significance is not indicated in Figure 4. Please indicate this significance (versus the 3rd cycle) or rephrase the sentence to explicitly include the baseline as the reference to which this difference is being compared.

33) Please replace "CD4+CD25+CD127" with "CD4+CD25+CD127-" (line 277).

34) Please replace "CTLs" using expanded wording as it is part of section title (line 287).

35) Please replace "of effector and memory/effector" with "effector and memory/effector cells" (line 309).

36) The authors claim that "However, in patients with lower OS, a tendency to decrease of effector and memory/effector, with a concomitant increase of naïve CD4+ T-cells, was detected" (line 309), however these tendencies seem not to reach statistical significance (Figure 9). Please indicate this fact in the text or indicate statistical significance by providing respective p-values in Figure 9.

37) Could the authors please better explain in the text what are the physiological consequences of the "tendency to decrease of effector and memory/effector with a concomitant increase of naïve CD4+ T-cells" (line 309)?

38) There are the two following formal issues with the sentence "Regarding CD8+ lymphocytes, a significant increase of cells with effector phenotype was detected in patients with higher OS" (line 311):

a) Despite the authors claim significantly increased levels of cells with effector phenotype in higher OS patients, p value is shown only for the 6th chemotherapy cycle in Figure 9. Please indicate statistical significance in Figure 9 or limit the finding to patients after 6 cycles of chemotherapy by rephrasing the sentence.

b) It is not clear to what condition is the increase being compared to? If the comparison is being made to lower OS patients, then it is strictly not precise to formulate this difference as an increase in high OS patients since this group is different from the low OS patients and therefore the amount of cells bearing the effector phenotype could not increase between them with time progression. Or do the authors mean that high OS patients displayed increased levels of cells with the effector phenotype?

39) Similarly, the sentence "In contrast, patients with lower OS showed a decrease in effector cell population and an increase of approximately 2-fold of naïve T-cells. These changes reached statistical significance" (line 312) is not semantically correct for the two analogous reasons:

a) Although statistical significance is claimed to be reached in low OS patients in terms of both the effector and naïve T-cell population changes, p values are displayed only for the 6th chemotherapy cycle. Please indicate statistical significance in Figure 9 or limit the conclusion to patients after 6 chemotherapy cycles by rephrasing the abovementioned statement.

b) Again, the formulation "patients with lower OS showed a decrease in effector cell population and an increase of approximately 2-fold of naïve T-cells" might suggest that there was an actual decrease or increase of effector cells or naïve T-cells, respectively, between the lower and higher OS patients, whereas it was actually the level of these immune cells that differed when comparing those two patient groups. Please specify precisely whether changes were observed between patient groups or occurred during the different phases of chemotherapy.

40) Please change "Variation according to treatment cycle in groups with shorter (n = 11) and longer (n = 19) overall survival are shown" to "Variations according to treatment cycle in groups with shorter (n = 11) and longer (n = 19) overall survival are shown" or "Variation according to treatment cycle in groups with shorter (n = 11) and longer (n = 19) overall survival is shown" (line 317).

41) The sentence "Total CD8+ T-379 lymphocytes and their effector subpopulations, identified by phenotypic markers and cytolytic proteins, decreased" (line 379) does not specify:

a) reference to which the changes in immune cell populations are being compared to

b) whether these changes occurred in lower or higher OS patients

42) The authors claim that "In this study eight cytokines and HMGB1 were quantified in patients with lung adenocarcinoma", however only TGF-beta and IL-6 seem to be shown in the results in addition to HMGB1 (Table 1, Figures 2, 3, 6). Please fix.

43) The sentence "Of the eight cytokines studied, five cytokines showed an increase" (line 395) is confusing for the following reasons:

a) the sentence seems to be incomplete as it is not clear whether the authors mean increase in cytokine levels or other parameter

b) it is not clear in relation to what reference these cytokines are being compared to

c) instead of 8, it looks like only 3 three cytokines (TGF-beta, IL-6, and HMGB1) were studied from the presented data

d) it is not clear which 5 cytokines actually showed an increase

44) Please replace "M. G-V." with "M.G-V." (line 410), "L. I-V." with "L.I-V. (line 412), "JS. L-G." with "JS.L-G.: (line 413).

45) Please change "biopsies" to "Biopsies" or "Biopsy" (line 410).

46) Please change "(CONACYT" to "(CONACYT)" (line 415).

Round 2

Reviewer 1 Report

The Authors have well addressed all my comments and there are no further requests from my side. The additionally provided information is well presented and certainly improves the publication. Therefore, I would like to congratulate the authors for their interesting manuscript.